# Genome-Wide Identification, Classification, and Expression Analyses of the *CsDGAT* Gene Family in *Cannabis sativa* L. and Their Response to Cold Treatment

**DOI:** 10.3390/ijms24044078

**Published:** 2023-02-17

**Authors:** Bowei Yan, Chuanyi Chang, Yingnan Gu, Nan Zheng, Yuyan Fang, Ming Zhang, Guijiang Wang, Liguo Zhang

**Affiliations:** 1Heilongjiang Academy of Agricultural Sciences Postdoctoral Programme, Institute of Industrial Crops, Heilongjiang Academy of Agricultural Sciences, Harbin 150086, China; 2Harbin Academy of Agricultural Science, Harbin 150028, China; 3Remote Sensing Technique Center, Heilongjiang Academy of Agricultural Sciences, Harbin 150086, China

**Keywords:** hemp, diacylglycerol acyltransferase (DGAT), gene family, expression patterns

## Abstract

Hempseed is a nutrient-rich natural resource, and high levels of hempseed oil accumulate within hemp seeds, consisting primarily of different triglycerides. Members of the diacylglycerol acyltransferase (DGAT) enzyme family play critical roles in catalyzing triacylglycerol biosynthesis in plants, often governing the rate-limiting step in this process. As such, this study was designed to characterize the *Cannabis sativa DGAT* (*CsDGAT*) gene family in detail. Genomic analyses of the *C. sativa* revealed 10 candidate *DGAT* genes that were classified into four families (*DGAT1*, *DGAT2*, *DGAT3*, *WS/DGAT*) based on the features of different isoforms. Members of the *CsDGAT* family were found to be associated with large numbers of cis-acting promoter elements, including plant response elements, plant hormone response elements, light response elements, and stress response elements, suggesting roles for these genes in key processes such as development, environmental adaptation, and abiotic stress responses. Profiling of these genes in various tissues and varieties revealed varying spatial patterns of *CsDGAT* expression dynamics and differences in expression among *C. sativa* varieties, suggesting that the members of this gene family likely play distinct functional regulatory functions *CsDGAT* genes were upregulated in response to cold stress, and significant differences in the mode of regulation were observed when comparing roots and leaves, indicating that *CsDGAT* genes may play positive roles as regulators of cold responses in hemp while also playing distinct roles in shaping the responses of different parts of hemp seedlings to cold exposure. These data provide a robust basis for further functional studies of this gene family, supporting future efforts to screen the significance of *CsDGAT* candidate genes to validate their functions to improve hempseed oil composition.

## 1. Introduction

Hemp (*Cannabis sativa* L.) is an annual herbaceous plant of the Cannabaceae family. As one of the oldest domestic crops, hemp has been widely used due to its industrial [1], ornamental [2], nutritional [3], and pharmaceutical [4] potential. Hempseed oil is a nutrient-rich edible oil with a lipid content of 25–35% and a unique fatty acid (FA) profile characterized by high levels of polyunsaturated fatty acids (PUFAs) and low levels of saturated FAs (SFAs) [5]. Depending on environmental and genetic factors, hempseed oil can consist of up to 90% unsaturated FAs, of which more than 70% are composed of PUFAs [6]. Hempseed oil is often regarded as a source of high levels of the essential fatty acids (EFAs) linoleic acid (18:2, n-6, LA) and α-linolenic acid (18:3, n-3, ALA), which are essential for human health and can only be obtained from dietary sources [6]. Hempseed oil also contains intermediate FAs such as γ-linolenic acid (18:3, n-6, GLA) and stearidonic acid (18:4, n-3, SDA) independent of precursor FAs [7]. These long-chain PUFAs (LCPUFA) are uniquely characteristic of hempseed, an ideal resource for stearidonic acid, which can serve as a precursor for the production of long-chained n-3 PUFAs [8]. LCPUFAs are physiologically essential regulators of neuronal development, inflammation, neurodegenerative activity, and cardiovascular health, yet they cannot be obtained from other common industrial oilseed crops.

Like most plant storage oils, hempseed oil primarily accumulates within the seeds, consisting mainly of triglycerides (TAGs) synthesized within the endoplasmic reticulum (ER) prior to accumulation in the form of lipid droplets (LDs) in the cytosol [9,10]. In oil crops, the primary mechanism that governs lipid assembly is the Kennedy pathway, which utilizes acyl-CoA as a substrate for fatty acyl residues and converts glycerol 3-phosphate to TAG in a four-step process [11]. The final step in this process, and the only rate-limiting step, is catalyzed by Acyl-CoA: Diacylglycerol Acyltransferase (DGAT, EC 2.3.1.20), which facilitates the esterification of a fatty acyl moiety to diacylglycerol (DAG) molecule to produce TAG, committing these molecules to the biosynthesis of TAG and shaping the carbon flow toward TAG production [12]. Therefore, diacylglycerol (DAG), the substrate of DGAT, accumulates during periods of rapid lipid formation in *Brassica napus* [13]. Generally, the *DGAT* gene family members can be divided into four subfamilies: the *DGAT1*, *DGAT2*, *DGAT3*, and Wax Ester Synthase *(WS)/DGAT*. The first *DGAT* gene in eukaryotes has been cloned in mice [14,15]. Members of the DGAT1 and DGAT2 subfamilies are transmembrane acyltransferases that localize to the ER with different membrane topologies and function as the primary isoenzymes to TAG synthesis in seed plants [14,15]. These two enzymes are thus likely to have evolved separately before undergoing functional convergence due to their similar acyltransferase activities utilizing DAG as a substrate [16]. Members of the DGAT3 subfamily are soluble cytosolic enzymes initially identified in peanut cotyledons [17]. Members of the dual-function WS/DGAT subfamily were initially characterized in *A. thaliana* and primarily catalyzed the biosynthesis of wax esters [18]. In addition, WS/DGAT enzymes catalyze the final step in producing small quantities of TAG, and their activity has been analyzed in many different plants [19].

The lipid accumulation process within plant tissues is complex and regulated by many genes [20]. Efforts to increase oilseed plant oil yields have spurred research focused on the identification and characterization of genes involved in the regulating of this agronomically valuable process so that breeders can seek to mutate or otherwise manipulate these genes to enhance oil quality [16]. Many studies of oil-rich plants, including *Glycine max*, *Arachis hypogaea*, *Brassica napus,* and *Vernicia fordii*, have emphasized the critical central role of reactions catalyzed by DGATs in the determination of the degree of TAG assembly and accumulation [21,22,23,24]. The transfer of the soybean *GmDGAT1-2* gene into Arabidopsis, for example, was associated with a 1.2-fold increase in total FA levels in the resultant transgenic plants relative to wild-type controls [25]. Overexpressing the peanut-derived *AhDGAT2a* gene in tobacco seeds impacted the transcription level of endogenous tobacco lipid metabolism genes and enhanced the FA content therein [26]. When the *BnDGAT* genes were overexpressed, this was sufficient to restore the synthesis of TAG in yeast H1246 while promoting FA accumulation therein [27].

Additionally, diverse environmental factors can impact plant oil formation, such that lipid-associated genes can govern blot growth and stress responses [28,29]. Plant lipid metabolic intermediates are important components of biofilm and active signal molecules, which are easily affected by environmental factors [30]. Low temperature is the leading environmental factor affecting crops’ geographical distribution, yield, and quality, which is closely related to fatty acids, protein, and other nutrients of hempseeds. Previous studies have shown that plants can respond to cold stress by accumulating lipids such as phosphatidic acid (PA), diacylglycerol (DAG), and triglyceride (TAG) in vivo [30]. DGAT is a vital substrate competitor for phospholipid and glycolipid synthesis, an important component of biofilm, so its coding genes play a crucial role in coordinating lipid metabolism [15]. It was shown that the conversion of phosphatidylcholine (PC) to triacylglycerol (TAG) was higher under cold treatment, and the expression of the *DGAT1* gene in cold-tolerant plants is higher than that in sensitive plants during a cold response, which promotes the accumulation of TAG in response to cold stress [31]. *A. thaliana* transforms DAG into TAG by overexpressing the *AtDGAT2* gene under cold stress to improve the plant’s cold tolerance [28]. The study on the cold response mode of the *DGAT* gene in hemp is of great value in improving the cold tolerance of hemp. However, it is unclear how many *DGAT* genes are contained in hemp, the function of its members, and the cold response pattern.

Hemp has long been cultivated and is a natural source of nutrients essential for optimal nutrition [32]. While DGAT enzymes are known to control the rate-limiting step in the process of lipid synthesis, the specifics of the copy numbers, sequences, structure, evolutionary relationships, and spatiotemporal expression patterns of Cs*DGAT* genes (*CsDGATs*) have yet to be analyzed comprehensively. Accordingly, the present study was constructed as a comprehensive bioinformatics analysis of the *CsDGAT* gene family. To aid efforts to improve the yield and quality of hempseed oil, we herein analyze the genomic architecture for different isoforms of the four classes of hemp *DGAT* genes (*DGAT1*, *DGAT2*, *DGAT3*, *WS/DGAT*) in comparison to those of other plant species. In addition, the gene structures, conserved domains, transmembrane domains, chromosomal positioning and collinearity, phylogenetic associations, subcellular localization, and *cis*-acting elements associated with these *CsDGATs* were characterized. Moreover, the expression of the identified *CsDGATs* in different tissues and response to cold stress conditions was evaluated using qPCR and transcriptomic approaches. Together, these findings offer new insight regarding the evolution of the *CsDGAT* gene family and the properties of these genes, thereby providing a basis for further studies of their biological functions in the context of *C. sativa* growth and development.

## 2. Results

### 2.1. Identification of C. sativa DGAT Gene Family Members

DGAT family proteins have been identified in various plant species [33]. To systematically identify candidate *DGAT* genes within the *C. sativa* genome, a hidden Markov model (HMM) and BLASTP methods were used together with whole-genome scanning. Following Pfam and Smart verification, redundant sequences in the annotated *C. sativa* genome were removed. The remaining 10 putative *CsDGAT* genes identified via this approach were tentatively designated *CsDGAT1*, *CsDGAT2*, *CsDGAT3*, *CsWSD1.1*, *CsWSD1.2*, *CsWSD1.3*, *CsWSD1.4*, *CsWSD1.5*, *CsWSD1.6*, and *CsWSD1.7* based on their subfamilies. The proteins encoded by these genes exhibited an average length of 480 amino acids, ranging from 327 aa (LOC115703123) to 556 aa (LOC115708250). The predicted molecular weights (MWs) of these proteins ranged from 37–62 kDa, while their predicted isoelectric point (pI) values ranged from 7.0–9.5, indicating that these proteins are likely to be alkaline (Table 1). Two online tools were used to predict the subcellular localization of the encoded proteins, suggesting that CsDGAT1 and CsDGAT2 would uniformly localize to the ER membrane or plasma membrane. In contrast, CsDGAT3 and different WS/DGAT family proteins were predicted to localize to diverse cellular compartments.

### 2.2. Phylogenetic Analyses of CsDGAT Gene Family Members

To understand the evolutionary relationships between *CsDGAT* genes and those of related plant species, a neighbor-joining (NJ) phylogenetic tree incorporating the protein sequences for 162 *DGAT* genes across 50 representative eukaryotic species, including eudicots, monocotyledons, ferns, and algae was constructed using MEGA X (Appendix A), with results being confirmed using a maximum likelihood (ML) tree. DGAT amino acid sequences yielded a well-resolved tree in which the four major DGAT subfamilies were separated from one another, including DGAT1, DGAT2, DGAT3, and WSD clades (Figure 1). Monocots and eudicots also form distinct clusters within these clades. Consistent with the tentative assignments shown above, CsDGAT1, CsDGAT2, CsDGAT3, and CsWSD proteins respectively clustered with the DGAT1, DGAT2, DGAT3, and WSD clades. The CsDGATs identified herein were more closely related to eudicot DGAT enzymes and were separated from those encoded by other species included in this analysis.

### 2.3. Chromosomal Distribution and Collinearity Analyses of CsDGAT Gene Family Members

The NCBI database was used to guide the mapping of these putative *CsDGAT* genes to the *C. sativa* genome, revealing their uneven distribution across five hemp chromosomes, with 1–5 genes per chromosome. Chromosome 1 encoded five of these *CsDGAT* genes (Figure 2A), while chromosomes 2, 5, and 6 only encoded a single *CsDGAT,* and the X chromosome encoded two *CsDGATs.* Notably, the number of *CsDGAT* genes per chromosome was not solely associated with chromosome size, given that the largest chromosome (Chr X) only encoded two *CsDGATs.* These *CsDGAT* genes were spread roughly equally across chromosomes 2, 5, and 9. These findings demonstrate that *CsDGAT* family genes were not evenly distributed across the five *C. sativa* chromosomes.

To more fully explore the evolution of these homologous *CsDGATs,* comparative syntenic maps were generated by comparing hemp to other representative species, including nine dicots (*Glycine max*, *Brassica napus*, *Arachis hypogaea*, *Arabidopsis thaliana, Ricinus communis*, *Sesamum indicum*, *Helianthus annuus*, *Gossypium darwinii*, *Juglans regia*) and three monocots (*Zea mays*, *Oryza sativa*, and *Sorghum bicolor*; Figure 2B). No duplicate pairs of genes resulting from tandem or segmental duplication were observed among these 10 *CsDGAT* genes. Overall, 19 *DGATs* exhibited collinearity relationships with identified *CsDGATs,* including three genes in *A. hypogaea*, three in *G. max*, three in *H. annuus*, four in *J. regia*, four in *R. communis*, and two in *S. indicum*. No such relationships were detected when comparing hemp sequences and those from the three representative monocots, consistent with a closer phylogenetic relationship with dicots than monocots (Figure 2B). These data suggest that members of the *CsDGAT* gene family may have the same function as collinear genes.

### 2.4. Analyses of CsDGAT Protein Structures and Conserved Domains

The genomic and protein-coding sequences of these *CsDGATs* were next utilized to conduct intron/exon structural analyses using the GSDS web tool. These analyses revealed distinct intron/exon structures for the members of each of these DGAT subfamilies (Figure 3A). Introns varied in length, with 2–16 introns per gene. *CsDGAT1* (LOC115704840) included the highest number of exons (n = 16), while *CsDGAT2* (LOC115703123) harbored nine exons, and *CsDGAT3* (LOC115722532) consisted of just two exons. The gene structures of the seven *CsWSD* genes identified herein, with most containing seven exons (*CsWSD1.1*, *1.2 1.3*, *1.4*, *1.5,* and *1.6*), while *CsWSD1.7* (LOC115702949) was comprised of 6 exons.

Proteins with highly conserved amino acid sequences, particularly in functional regions, often exhibit similar biological functions. Accordingly, the MEME software tool was employed to analyze the amino acid sequence domains of these putative CsDGATs, with five additional plant species also having been retrieved and the maximum number of motifs per protein sequence set to six. Orthologous sequences were identified among these six analyzed plant species, including six DGAT1 sequences, six DGAT2 sequences, six DGAT3 sequences, and 12 WS/DGAT sequences (Appendix A). Conserved motifs were then analyzed in these putative DGAT proteins, revealing the presence of shared protein motifs across the most highly conserved species (Appendix A).

The NCBI Conserved Domain Database and DNAMAN software were additionally used to characterize the structures of these DGAT proteins. This approach revealed that CsDGAT1 harbored a C-terminal MBOAT (PF03062) domain between amino acids 177 and 533 and nine conserved transmembrane regions that reaffirm its identity as a membrane-bound O-acyl transferase (Figure 3B). Multiple sequence alignment additionally demonstrated the presence of an Acyl-CoA binding signature (R^135^-G^153^), a putative catalytic active site (R^168^LIIEN^173^), a phosphopantetheine attachment site (G^177^ to M^199^), an SnRK1 target site (F^217^-X-X-X-X-X-I^223^-X-X-X-VV^228^), a putative thiolase acyl-enzyme intermediate signature (C^242^-P-XX-V-X-L-R-X-DSA-X-LSG-XX-L-XXX-A^264^), a putative fatty acid protein signature (A^408^E^409^-X-L-X-FGDREFY-X-DWWN^424^), a DAG-binding motif (HRW-XX-RH-X-Y-X-P) and a putative C-terminal ER retrieval motif (-YYHDV-) domain in this protein (Appendix A).

CsDGAT2 was found to harbor a PlsC domain located between amino acids 124 and 232, classifying it as a lysophospholipid acyltransferases (LPLAT) superfamily member (PF03982), with two N-terminal transmembrane domains also being identified in this analysis (Figure 3B). Alignment analyses conducted for 70 species of plants, animals, and microbes revealed six highly conserved domains in members of the *DGAT* gene family [34]. Here, CsDGAT2 sequence alignment with the DGAT2 proteins encoded by Arabidopsis (AT3G51520), peanut (AEO11788), and rice (Os02g48350), soybean (Glyma01G156000), and maize (GRMZM2G050641) was performed, revealing conserved PH Block, PR Block, GGE Block, RGFA Block, VPFG Block, and G Block motifs, in line with prior research evidence (Appendix A).

CsDGAT3 was not found to encode any transmembrane domains and contained a single TRX domain (Figure 3B). The comparison of the full CsDGAT3 sequence with sequences of other known acyltransferases confirmed that a phosphopantetheine attachment site was present in CsDGAT3 at S^24^-G^38^, with a putative thiolase acyl-enzyme intermediate signature S^166^XXXXXXSXXS^176^ also being detected therein (Appendix A). CsDGAT3 also harbored a fatty acid protein binding signature (KSGSAALVEEFERVMGAE), and a putative active catalytic site was detected between the NLFRDE residues.

All CsWSD proteins were found to harbor conserved N-terminal WES (PF03007) domains and conserved C-terminal DFU (PF06974) domains (Figure 3B). Some of these CsWSD proteins contain transmembrane regions and other conserved domains. CsWSD1.2, CsWSD1.6, and CsWSD1.7, for example, each contained 1–2 transmembrane regions, while CsWSD1.3 harbors an AAtase domain (PF07247). Different members of these *CsDGAT* gene subfamilies may have thus evolved distinct functions. Multiple sequence alignment was also proposed for a proposed N-terminal active-site motif (HHXXXDG), revealing its presence in these candidate WS/DGAT1 coding sequences (Appendix A).

### 2.5. CsDGAT Secondary and Tertiary Structural Analyses

The structures of these 10 CsDGAT proteins were compared in greater detail, with SOPMA being used to examine their predicted secondary structural characteristics. These CsDGATs included α-helix, extended strand, beta-turn, and random coil elements (Appendix A), with over 70% of the secondary structure comprising α-helix and random coil elements. In contrast, the proportion of extended chain and beta-turn elements in the overall CsDGAT protein sequences was relatively low (Appendix A).

A homology modeling method and the SWISS-MODEL database were further used to construct the predicted 3D structures of these CsDGAT proteins, with the optimal structure being that with the highest score. Highly significant differences were observed when comparing the 3D structures of members of these different CsDGAT subfamilies (Appendix A), with CsDGAT1 and CsWSD1 subfamily members being the most structurally complex. The diverse structural characteristics of these CsDGATs provide a valuable foundation for further research to clarify their physiological functions.

### 2.6. Identification of CsDGAT Promoter cis-Acting Elements

*Cis*-acting regulatory elements are short sequences within the promoter regions upstream of particular genes that can be recognized and bound by specific transcription factors, thereby regulating gene expression. The 2000 bp promoter regions upstream of each *CsDGAT* gene were next analyzed to identify predicted *cis*-acting elements. This approach led to the identification of 46 such elements related to stress responses, light responses, phytohormone regulation, and plant development (Figure 4). The *CsDGAT* promoter regions contained many lights and phytohormone response elements, including 11 G-box elements and seven ABRE elements in the *CsDGAT3* promoter. These analyzed promoters also detected many other stress-response elements were also detected in these analyzed promoters, including cold-responsive, defense, and stress-responsive *cis*-acting elements. These findings emphasize the potential involvement of different *CsDGAT* genes in hemp plant development, environmental adaptation, and abiotic stress responses.

### 2.7. CsDGAT Subcellular Localization Analyses

Full-length CsDGAT sequences for six representative genes (CsDGAT1, CsDGAT2, CsDGAT3, CsWSD1.1, and CsWSD1.4) were fused to a GFP reporter gene under the control of the 35S promoter and introduced into *N. benthamiana* leaves to validate these predicted localization results. High levels of GFP expression were observed in the cytosol, membrane, and nuclei of the cells of these tobacco leaves. Specifically, the CsDGAT1-GFP, CsDGAT2-GFP, and CsWSD1.1-GFP fusion proteins localized to the ER, while CsDGAT3-GFP localized to the chloroplast compartment, and CsWSD1.4-GFP was present in the nuclei and cytosol, largely consistent with the above predictions (Figure 5).

### 2.8. CsDGAT Gene Expression Patterns in C. sativa

To explore patterns of *CsDGAT* expression in *C. sativa*, RNA-seq data corresponding to female inflorescences from nine varieties of hemp and five different tissues from the Longdama 6 hemp variety (roots, stems, leaves, seeds, flowers) were used to construct gene expression heat maps and cluster diagrams (Figure 6, Appendix A). The *CsDGAT1*, *CsDGAT2*, *CsDGAT3*, *CsWSD1.2,* and *CsWSD1.7* expression varied markedly across these hemp varieties, with *CsDGAT1, CsDGAT3,* and *CsWSD1.2* being expressed at relatively high levels. However, the other analyzed *CsDGATs* were largely undetectable in these different female inflorescences (Figure 6A). *CsDGATs* were also expressed in tissue-specific patterns in a given hemp variety, with all 10 genes being expressed in at least one of the analyzed tissue types (Figure 6B). While *CsDGAT1*, *CsDGAT2*, and *CsWSD1.2* were expressed at high levels in seed and flower tissue samples, *CsWSD1.1* and *CsWSD1.7* were specifically expressed in the stem, leaf, and root samples, and *CsWSD1.3* was expressed at a higher level in leaves and stems. The expression of *CsWSD1.4* was only detected in seed tissue samples. *CsDGAT3* expression levels tended to be high across various tissues, suggesting a role for this gene in the process of tissue development. *CsWSD1.5* and *CsWSD1.6* were expressed at relatively low levels in all analyzed tissues, suggesting that these genes may not be functional or that they exhibit spatiotemporally-specific expression patterns, which are unlikely to be important regulators of hemp development. Accordingly, different *CsDGATs* are likely to play distinct and varied roles in the growth and development of hemp plants. In addition, the analysis of the expression patterns of *CsDGAT* genes in different seed development stages showed that the expression of *CsDGAT3* and *CsWSD1.4* genes were higher in the whole seed development stage, which was significantly higher than that of other genes, on the contrary, the expression levels of *CsWSD1.3*, *CsWSD1.5,* and *CsWSD1.6* were lower than those of other genes. *CsDGAT1* and *CsDGAT3* were higher in the early and later stages of seed development, and *CsWSD1.2* and *CsWSD1.7* were higher in the middle stage of seed development. The expression of *CsWSD1.4* is relatively high in the middle and later stages of seed development. It is worth noting that the expression of *CsDGAT2* maintains a relatively stable expression level in the whole stage of seed development, indicating that *CsDGAT* family genes function in different stages of seed development. It also predicts that in higher plants the process of lipid accumulation process is finely and strictly regulated.

The expression patterns of these 10 *CsDGAT* genes were validated in different tissues via qPCR (Figure 7). This approach revealed patterns of *CsDGAT* expression in different tissues of the Longdama 6 hemp variety that differed somewhat from the above transcriptomic data trends. However, the expression of these genes tended to be higher in seeds and female flowers relative to other tissues. Relatively high levels of *CsDGAT3*, *CsWSD1.3*, *CsWSD1.6*, and *CsWSD1.7* expression were observed in leaves, while marked differences in *CsWSD1.4* expression were observed across tissue such that it was only expressed at high levels in seeds.

### 2.9. Phenotype and CsDGAT Expression Pattern Analysis under Cold Stress Response in C. sativa

Under cold stress, the leaves of hemp plants showed symptoms of wilting and water loss, and the degree of gradual wilting deepened with the duration of stress (Figure 8A). The morphology of the plant without cold stress was normal, and the degree of leaf wilting was not evident under short-term cold stress for 12 h; the degree of leaf wilting began to be obvious at 24 h, and the leaves of hemp plant wilted at 48 h. Severe wilting was observed after 72 h of cold stress (Figure 8A(a)). From the leaf morphology at different treatment times, it can be seen that the degree of curling of hemp leaves increased with the extension of treatment time, indicating that cold stress led to changes in plant morphology (Figure 8A(b)). *CsDGAT* gene expression patterns were next examined in the roots and leaves of *C. sativa* seedlings exposed to cold stress. Under these cold conditions, *CsDGAT1, CsDGAT2,* and *CsDGAT3* genes were significantly (>2-fold) upregulated in the leaves of these seedlings in the middle and late stage of processing relative to untreated plants (Figure 8B,C). In contrast, all *CsWSDs* other than *CsWSD1.2*, *CsWSD1.4,* and *CsWSD1.5* were slightly downregulated relative to control conditions. The *CsDGAT* expression patterns observed in *C. sativa* roots were distinct from those observed in leaves, with all of these genes other than *CsWSD1.7* being upregulated in response to cold stress at different time points. Specifically, *CsDGAT1, CsDGAT2,* and *CsDGAT3* were significantly upregulated in the middle and late stages of processing, with consistent patterns in both root and leaf tissues relative to control samples. In contrast, *CsWSD1.1*, *CsWSD1.4*, *CsWSD1.5*, and *CsWSD1.6* were initially downregulated and then upregulated under cold stress conditions, suggesting that the biological clock of these hemp plants may at least partially control the expression of certain *CsDGATs*.

## 3. Discussion

Members of the *DGAT* gene family were initially characterized over two decades ago, and the enzymes encoded by these genes play a central role in vegetable oil production, metabolic regulation, and stress response leading to their widespread study in a range of plant species. They are also the only rate-limiting enzymes involved in the Kennedy pathway that governs de novo TAG biosynthesis [15,29]. In contrast to the *DGAT* gene families of the primary oilseed plant species, however, the hemp *DGAT* family remains to be characterized in detail. A growing number of studies have shown that *DGAT* genes are important targets for efforts to understand better the mechanisms that govern lipid metabolism and regulation [35]. The publication of the *C. sativa* genome has provided an opportunity to systematically characterize the key functional genes encoded therein to guide molecular breeding efforts [36]. Thus, there is a clear need to more fully explore *CsDGAT* gene resources to understand and regulate TAG biosynthesis in this economically important species and to provide a theoretical basis for elucidating the molecular mechanism of the key *CsDGAT* gene of lipid metabolism regulation pathway in the cold response of industrial hemp.

### 3.1. CsDGAT Gene Family Identification

This study identified 10 putative *DGAT* genes in the *C. sativa* genome, including one *DGAT1,* one *DGAT2,* one *DGAT3,* and seven *WSD1* genes. Exon-intron structural diversification analyses are critical to developing a detailed understanding of the evolution of particular gene families across species [37]. In this study, the divergent exon-intron organization was observed among the four *CsDGAT* gene families, consistently exhibiting a unique evolutionary history and emphasizing the diversity of plant DGATs [14,37]. Marked differences in the length and distribution of intronic and UTR sequences were a major contributor to observed gene structure diversification [14]. Generally, only limited amino acid sequence similarity was observed between *CsDGAT* genes in these four different subfamilies, suggesting that they play distinct and nonredundant roles in regulating hemp physiology.

The functional domains of *CsDGAT* gene family members were additionally characterized, revealing conserved MBOAT (PF03062), LPLAT (PF03982), WES (PF03007), and DFU (PF06974) domains. Conserved sequences and active catalytic sites in these DGAT homologs were also analyzed, revealing that these proteins shared very low levels of identity with members of other CsDGAT subfamilies (Appendix A). Gene synteny analyses indicated an absence of synteny among the four CsDGAT subfamilies (Figure 2), and significant differences in intron/exon gene structure were observed among these four subfamilies (Figure 3A). Conserved domains and motifs were detected in members of all four analyzed DGAT subfamilies (Figure 3B and Appendix A), in line with prior reports studying peanut [22] and oil palm plants [16]. While some variations in sequence homology were noted among different *DGAT* genes encoded by the same species, the conserved domains and functional motifs are likely to be associated with active sites that are important for substrate binding, catalytic activity, and other key regulatory roles [16].

There has been a growing research focus on the role of *cis*-acting elements that influence gene expression in recent years, with many of these elements being responsive to environmental stressors and phases of plant development [38]. A promoter analysis for the *CsDGATs* identified herein revealed several stress-responsive, light-responsive, phytohormone-responsive, and plant development-related *cis*-acting elements, emphasizing the potential interactions between these hormone-responsive genes and a range of other metabolic pathways. Throughout evolution, plants have developed a range of unique approaches to adapting or responding to external environmental factors. DGAT family members have been documented as central mediators of responses to biotic and abiotic stressors, including saline-alkali, cold, and drought stress conditions [29].

### 3.2. Phylogenetic Relationships among DGAT Family Proteins

Phylogenetic and syntenic analyses of DGATs encoded by different species have provided detailed insight into their evolution and function [14,16,29]. Members of the *DGAT1* and *DGAT3* gene families have convincingly been shown as originating from different ancestors during the emergence of eukaryotic species and to have evolved in a non-symmetric manner through convergent evolution [14,24]. *DGAT3* and *WS/DGAT* family genes also form a monophyletic subfamily that is phylogenetically divergent from the DGAT1 and DGAT2 subfamily [14], with these genes being primarily encoded by plants and largely absent in animal species [14,24]. In contrast, *DGAT1* and *DGAT2* are widely expressed in plants, animals, and other major eukaryotic species. The specific patterns of genes in the *DGAT1*, *DGAT2*, *DGAT3*, and *WS/DGAT* subfamilies also differ from one another. In hemp, the *WS/DGAT* gene was the most diversified, with seven *CsWSDs* having been identified in the present analysis; in contrast, Cs*DGAT1*, Cs*DGAT2,* and Cs*DGAT3* were maintained as single copies, while multiple genes have been reported in the *DGAT1* and *DGAT2* subfamilies in many plants [14,16]. CsDGATs were most closely related to DGATs encoded by eudicots in phylogenetic analyses, indicating that they may have diverged early during the process of plant evolution, originating prior to plant diversification with DGAT activity having remained largely unchanged owing to the importance of TAGs in all species [14].

### 3.3. Profiling of CsDGAT Expression across Hemp Tissues and Varieties

Gene expression patterns are closely tied to the functional roles of these genes in plants [39]. DGAT enzymes are generally considered to play essential roles in the process of TAG biosynthesis [40], underscoring the value of studying these genes given the universal importance of TAGs across the various domains of life. Different *DGAT* genes have been shown to exhibit varying expression profiles across plant species, consistent with their species-specific functional roles [23]. Evaluating the expression profile for a particular gene across tissues can offer a foundation for subsequent research [16]. Accordingly, the expression of the putative hemp *CsDGAT* genes identified herein were compared through in silico and qPCR approaches, revealing gene-specific expression profiles. *CsDGAT1*, *CsDGAT2*, *CsDGAT3*, *CsWSD1.2,* and *CsWSD1.7* were expressed at high levels in female inflorescences from various hemp varieties, and they may play a key role in floral development. *AtDAGT1* and *AhDAGT1* have previously been reported to be expressed at the highest levels in flower and seed tissues. Tissue-specific *CsDGAT* expression patterns were also observed within a given hemp variety, with all *CsDGATs* being expressed in at least one tissue compartment. For example, *CsDGAT1*, *CsDGAT2,* and *CsWSD1.2* were expressed at high levels in seed and flower tissues, consistent with their potential functional roles in seed or flower development, while *CsDGAT3* was highly expressed in many tissues suggesting it is more important in the context of general tissue development. *CsWSD1.5* and *CsWSD1.6* were expressed at relatively low levels across analyzed tissues, suggesting that they may not play major roles in the growth or development of the analyzed hemp variety.

Different patterns of *DGAT* gene expression were also observed across species. For example, the expression of *DGAT3* in Arabidopsis is evident at high levels across different stages of development, with >2-fold upregulation evident during early seed development [41]. In contrast, peanut *DGAT3* was expressed at the lowest levels in seeds and the highest in leaves and flowers [22]. *CsDGAT* gene expression also varies across hemp varieties, highlighting probable roles for these genes as regulators of gene expression and key functional processes. Further study of the distinct spatial expression patterns of members of the same classes of *DGAT* genes in different plants is warranted. Generally speaking, in oilseeds rich in normal fatty acids, such as Arabidopsis, soybean, and rapeseed, *DGAT1* shows much higher expression levels than *DGAT2* in developing seeds, and therefore, DGAT1 is the main enzyme for TAG synthesis in these oilseed plants [42]. However, *DGAT2* is expressed at higher levels during seed development in some plant species capable of accumulating unusual fatty acids, such as *Vernicia fordii* [43], *Cyperus esculentus* [44], and *Ricinus communis* [45], which have a higher affinity for unusual fatty acid -containing substrates. The expression of the *CsDGAT2* gene was observed to be higher than *CsDGAT1* in this study, and it is noteworthy that the expression of the *CsDGAT3* gene was higher in female flowers of different species, in different tissue parts and in different developmental periods of seeds, which may be associated with the formation of some specific fatty acids among hempseeds. However, the relative contribution and potential regulatory mechanism of *CsDGAT* genes of different family members to TAG synthesis need to be determined by more biochemical and molecular biological analysis.

### 3.4. The Response of CsDGAT Genes to Cold Stress

Cold stress is one of the primary environmental factors that can adversely impact the quality and yield of specific crops. Plants engage in a series of physiological and biochemical processes to adapt to cold and other abiotic stressors. Structural changes in plant cells are one of the important mechanisms by which the cell membrane system suffers first damage when plants are subjected to cold stress. It has been reported that membrane lipid composition and unsaturation are closely related to plant cold tolerance [46]. Triacylglycerol (TAG) is the main component of plant oil and an important donor source of cell membrane lipid synthesis in plants [47]. As a key enzyme for TAG synthesis, many studies have supported a potential role for *CsDGAT* family members in plants’ cold responses [29]. Arabidopsis *DGAT1* gene was induced by low-temperature stress to upregulate expression [28], and the *DGAT2* gene can be induced by cold, heat, and some biotic stresses in soybean and peanut [22]. Additionally, following exposure to cold, the galactolipid MGDG undergoes conversion into DAG and oligogalactolipids, potentially further catalyzing TAG production in a manner conducive to membrane stability [29]. In this study, all 10 identified *CsDGAT* genes were significantly responsive to cold stress, tending to be upregulated at different time points while exhibiting distinct spatial regulatory patterns when comparing leaves and roots that suggest a possible role for *CsDGAT* genes as positive regulators of cold responses in hemp. Under cold stress, maize leaves were previously shown to exhibit the upregulation of 5 *ZmDGAT* genes, with such upregulation occurring later following cold treatment and more strongly than similar *ZmDGAT* gene induction observed in maize roots [29]. Root *CsWSD* gene responses, in contrast, were stronger than those in leaves. As such, these genes are likely to play distinct roles in different parts of hemp seedlings when responding to cold stress. In this study, the role and regulatory mode of *CsDGAT* gene family members in cold tolerance of hemp were initially investigated, which laid the foundation for the subsequent study of the function of the *CsDGAT* gene family and the selection of new cold-tolerant hemp varieties using molecular breeding methods.

In order to analyze the function of the *CsDGAT* gene more comprehensively, our team will construct the expression vector of the target gene promoter fused with the GUS tag for Arabidopsis transformation and analyze the expression pattern of the target gene by tissue staining method. To further validate the cold tolerance function of the *CsDGAT* gene, our team will continue to construct yeast expression vectors and verify the enzymatic and cold response function of the target gene in yeast lipid synthesis mutant *H1246*. Furthermore, the target gene will be over-expressed in Arabidopsis mutant *AS11* and wild type, and the comprehensive methods of biochemistry, bioinformatics, molecular biology, and multi-group association analysis will be used to validate the lipid synthesis and cold reaction function of the target gene in plant models. The molecular mechanisms and signaling pathways involved in the cold response of the *CsDGAT* gene in hemp will be explained more comprehensively.

## 4. Materials and Methods

### 4.1. CsDGAT Gene Identification

Hemp genome sequence information was downloaded from the NCBI database (GenBank assembly accession: GCA_900626175.2). Arabidopsis *DGAT* genes were utilized as a reference for the identification of orthologous *CsDGAT* gene family members. To identify other *CsDGATs,* particularly singleton genes, a hidden Markov model (HMM) profile was constructed from orthologous amino acid sequences and used as an HMM search query against hemp gene models using the HMMER3 package [48,49]. To identify additional hemp *DGAT* genes, BLASTP was used for sequence similarity searches against a reference sequence protein database [50]. ExPASy (https://web.expasy.org/protparam/, last accessed: 14 May 2022) was used to compute the isoelectric point and molecular weight of putative CsDGAT proteins identified herein. The subcellular localization of the 10 identified CsDGATs was analyzed with the Softberry (http://linux1.softberry.com/, last accessed: 14 May 2022) and CELLO (http://cello.life.nctu.edu.tw/, last accessed: 14 May 2022) tools [51].

### 4.2. Sequence Similarity and Phylogenetic Analyses

The amino acid sequences of DGATs in a range of representative eukaryotic plant species, including monocotyledons, eudicots, fern, mosses, and algae, were downloaded from the NCBI and Phytozome 12.0 database (Appendix A) [29]. These DGAT sequences and the *CsDGAT* sequences identified herein were aligned using ClusalW and used to construct a neighbor-joining tree with 1000 bootstrap replicates using MEGA 7.0 [52].

### 4.3. Chromosomal Distribution and Collinearity Analyses

The locations of putative *CsDGAT* genes within the *C. sativa* genome were determined based on genomic sequence and annotation files, followed by mapping to appropriate chromosomes. Collinearity analyses were performed by comparing orthologous genes in *C. sativa, Zea mays (GCF_902167145.1)*, *Oryza sativa* (GCF_001433935.1), *Sorghum bicolor* (GCF_000003195.3_), *Arabidopsis thaliana* (GCF_000001735.4), *Gossypium darwinii* (GCA_013677245.1), *Glycine max* (GCF_000004515.6), *Brassica napus* (GCF_000686985.2), *Arachis hypogaea* (GCF_003086295.2), *Ricinus communis* (GCF_000151685.1), *Sesamum indicum* (GCF_000512975.1), *Helianthus annuus* (GCF_002127325.2), and *Juglans regia* (GCF_001411555.2) with the TBtools MCScanX toolkit [53].

### 4.4. Analyses of Gene Structure and Conserved Protein Domains

*CsDGAT* gene exon-intron structures were displayed with the Gene Structure Display Server (GSDS) [54]. NCBI CDD (https://www.ncbi.nlm.nih.gov/ Structure/cdd/wrpsb.cgi last accessed: 14 May 2022) and SMART (http://smart.embl-heidelberg.de/ last accessed: 14 May 2022) were used to identify conserved protein domains. *DGAT* gene structures were generated with the IBS tools [55]. Conserved amino acid sequences for *CsDGATs* were identified using MEME SUITE (minimum width ≥ 6, maximum width = 50) [56]. The Transmembrane Prediction Tool (https://services.healthtech.dtu.dk/service.php?TMHMM-2.0, last accessed: 14 May 2022) was used to predict transmembrane domains. DNAMAN tools were used for cluster members’ multiple sequence alignment and visualization of cluster members. SOPMA (http:///npsapbil.ibcp.fr/, last accessed: 15 September 2021) was utilized to assess the inferred secondary structure. Tertiary structural predictions were made using SWISS-MODEL (https://swissmodel.expasy.org/, last accessed: 15 June 2022).

### 4.5. Cis-Acting Element Analyses

The *C. sativa* genome was queried to identify the promoter sequences (defined as the 2 Kb region upstream of the start codon) for each of the identified *CsDGAT* promoter sequences, and the numbers and types of *cis*-acting elements in these promoters were analyzed with the Online PlantCARE software program (http://bioinformatics.psb.ugent.be/webtools/plantcare/html/, last accessed: 15 June 2022) [57]. Results were then visualized with PowerPoint, GraphPad Prism, and TBtools [53].

### 4.6. Subcellular Localization Analyses

The Primer 5.0 software was used to design primers, including enzyme cleavage sites (Appendix A), after which the full-length coding sequence (CDS) of each candidate gene was cloned into the pBWA(V)HS-GFP vector. The resultant pBWA(V)HS-*CsDGATs*-GFP plasmids were introduced into *Agrobacterium tumefaciens* (EHA105) via electroporation, with empty plasmids serving as a negative control. Monoclonal cells that were positive for successful transformation were cultured in liquid LB medium containing kanamycin (50 mg/L) and were collected via centrifugation, after which bacteria were suspended in a solution containing 10 mM MES (pH 5.7), 10 mM MgCl_2_, and 150 μM acetosyringone. The suspension was cultured in the dark to an OD_600nm_ of 0.8–1.0. Then, this inoculum was used to infiltrate the lower epidermis of 4-week-old *Nicotiana benthamiana* leaves using a needleless syringe until the leaves had been fully penetrated. These tobacco leaves were cultivated in the dark for 3 days, after which a confocal microscope (Nikon C2-ER) was used to visualize the GFP fluorescent signal.

### 4.7. Analyses of CsDGAT Expression Profiles in Different Tissues and Varieties

RNA-seq data were downloaded from the NCBI (PRJNA498707) corresponding to the female inflorescences of 9 different hemp varieties, including the Mama Thai (MT), White Cookies (WC), Canna Tsu (CT), BlackLime (BL), Temple (T), Cherry Chem (CC), BlackBerry Kush (BB), Sour Diesel (SD), and Valley Fire (VF) varieties. Meanwhile, transcriptomic data corresponding to the roots, stems, leaves, flowers, and seeds of the ‘Longdama No. 9’ hemp variety were generated by our team, and the RNA libraries were sequenced on the Illumina NovaseqTM 6000 platform by OE Biotech, Inc., Shanghai, China (PRJNA899681). In addition, transcriptome data from hemp seeds at different developmental periods (10d, 20d, 27d) after fertilization were used to analyze the expression profiles of *CsDGAT* genes at different developmental stages of hempseeds (PRJNA513221). Heat maps were constructed using TBtools. In addition, samples of the root, stem, leaf, flower, and seed tissues from female hemp plants were collected in October 2021 and snap-frozen with liquid nitrogen for subsequent qPCR analysis.

### 4.8. Cold Treatment and qPCR

Cold treatment and gene expression analyses were performed using the ‘Longdama No. 9’ hemp cultivar, breeding from the Industrial Hemp Group of the Institute of Industrial Crops (Heilongjiang Academy of Agricultural Science). All hemp seedlings were grown in a greenhouse at 25 °C under an artificial light source providing 16 h of light and 8 h of dark per day. Seedlings were grown until exhibiting four pairs of true leaves, at which time they were exposed to cold (4 °C). Leaves were collected from these plants at baseline before cold exposure (0 h) and at 12, 24, 48, and 72 h of cold exposure, with three repeat experiments (n = 3 plants/experiment) for each treatment. All samples were snap-frozen with liquid nitrogen and stored at −80 °C.

An RNA extraction kit (Omega Bio-Tek, Shanghai, China) was used to extract total RNA from these frozen samples, after which an EasyScript^®^ One-Step gDNA Removal and cDNA Synthesis SuperMix (Transgen, Beijing, China) were used to prepare cDNA based on provided directions. The TransStart^®^ Top Green qPCR SuperMix (Transgen, Beijing, China) was then used to perform qPCR based on provided directions. Actin served as a reference gene, and the comparative 2^−ΔΔCT^ method was used to assess relative gene expression. Three biological replicates and three technical replicates were included in all experiments. Primers used for this analysis are provided in Appendix A.

### 4.9. Statistical Analysis

GraphPad Prism 8.0 software was used for one-way ANOVA, and Duncan’s test was used for multiple comparisons and significance of differences analysis, where the graphical data were the mean of three or more replications.

## 5. Conclusions

In the present study, the *CsDGAT* gene family was herein analyzed, identifying 10 candidate *CsDGAT* genes classified into four subfamilies. In summary, a comprehensive and systematic bioinformatics analysis was carried out on the members of the *CsDGAT* family, including homologous relationships, chromosomal locations, collinearity, conserved amino acids, gene structures, conserved motifs, evolutionary relationships, and *cis*-acting elements associated with these *CsDGAT* family members, in order to understand their potential gene functions better. To further understand how they may regulate hemp growth and development, the subcellular localization and the expression patterns of these *CsDGATs* were also analyzed. The results showed that most of the genes have different expression patterns in different varieties, tissue parts, and seed development stages, indicating that their functions are space-time differences in their functions. In addition, the cold response pattern of the *CsDGATs* gene was analyzed, which showed that it had different expression patterns in leaf and root tissues. Together, these results offer a valuable foundation for further studies of the functional roles of *CsDGAT* genes, enabling future functional validation studies while supporting efforts to improve hempseed oil composition and cold resistance.

## Figures and Tables

**Figure 1 ijms-24-04078-f001:**
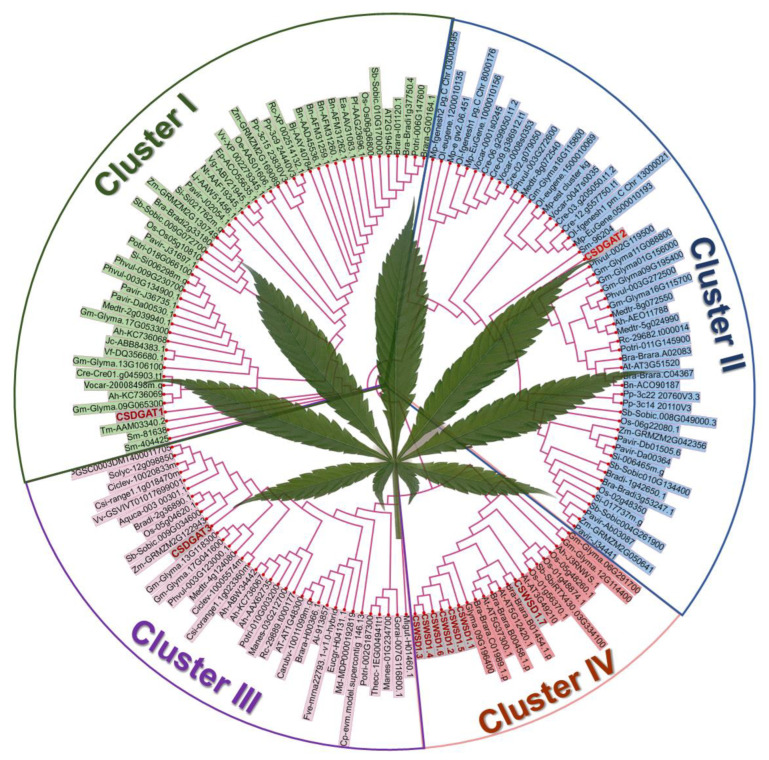
Phylogenetic analysis of *CsDGAT* gene family. A phylogenetic tree of CsDGAT proteins from *C. sativa* and other plants was constructed by MEGA 10.0 using full-length protein sequences. Different branches were distinguished with different color shades. CsDGAT proteins were represented by red color. For the description of other species abbreviations involved in the figure, please see Additional Appendix A.

**Figure 2 ijms-24-04078-f002:**
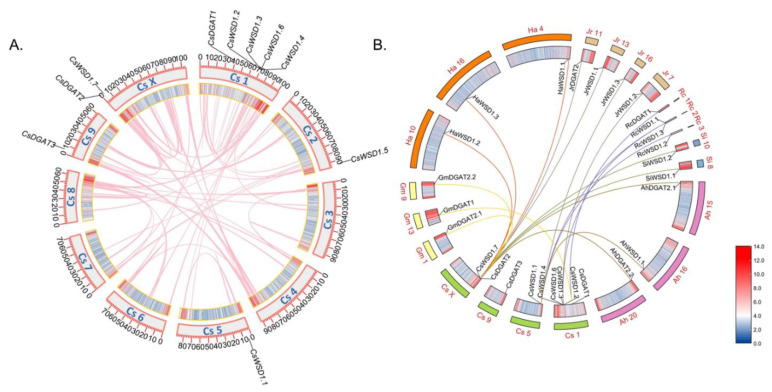
Chromosomal location and synteny analysis of *CsDGATs* in *C. sativa* genome. (**A**) Chromosomal locations of *CsDGATs*. Tandem-duplicated genes are indicated with red boxes, and the chromosome number is indicated above each chromosome. The scale is in megabases (Mb). (**B**) Syntenic relationship of *CsDGATs*. The annotations on the fragments represent different chromosomes, and the numbers in the outermost circle represent the positions of the corresponding chromosomes. The *CsDGATs* involved in segmental duplications in the *CsDGATs* gene family are mapped to their respective locations of the *C. sativa* genome in the circular diagram. The different color lines represent the segmental duplication pairs between the *CsDGATs* and the *DGAT* gene of other plant genomes. *Gm*, *Glycine max* (yellow font), *Ha*, *Helianthus annuus* (orange font), *Jr*, *Juglans regia* (grey font), *Rc*, *Ricinus communis* (blue font), *Si*, *Sesamum indicum* (green font), *Ah*, *Arachis hypogaea* (brown font).

**Figure 3 ijms-24-04078-f003:**
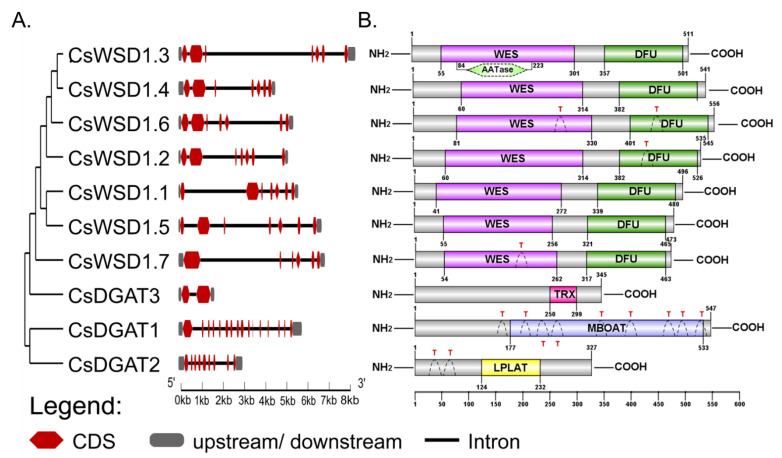
Structures for the *CsDGAT* gene family. (**A**) Phylogenetic relationships and gene structures for the *CsDGAT* gene family. Phylogenetic tree generated by the neighbor-joining tree method with bootstrapping analysis (1000 replicates) based on the protein sequences of CsDGAT. The exon-intron structures for *CsDGAT* gene family members were visualized with GSDS2.0. The horizontal black lines, red boxes, and gray boxes show introns, coding sequence, and untranslated region, and the scale displays the relative length and position of the introns and exons. (**B**) Model illustrating transmembrane regions and putative conserved domain structure of 10 CsDGAT protein identified in *C. sativa*. The CsDGAT family can be divided into three subgroups based on the different conserved domains: CsDGAT1, CsDGAT2, CsDGAT3, and CsWSD1. The grey bars represent the length of each protein sequence, and conserved domains are shown as colored boxes. The conserved domain architectures prediction relies on SMART, and IBS software was used for visualization with default parameters. MBOAT = membrane-bound O-acyl transferase domain, LPAT = lysophospholipid acyltransferase domain, TRX = thioredoxin-like, ferredoxin family domain, WES = wax ester synthase-like Acyl-CoA acyltransferase domain, DUF = domain of unknown function, AATase = alcohol acetyltransferase domain, T: transmembrane domains; The numbered bar indicates the position of amino acid.

**Figure 4 ijms-24-04078-f004:**
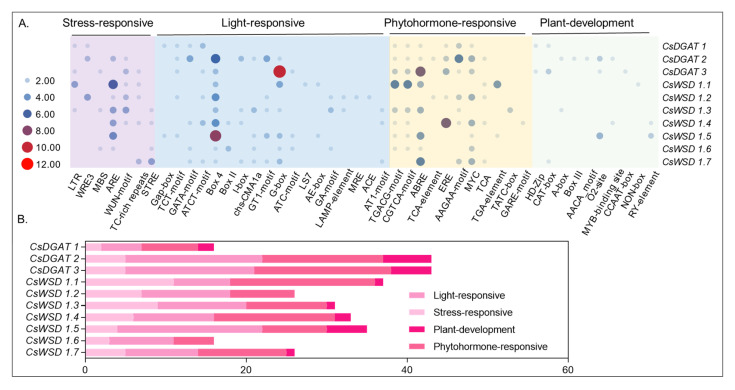
The schematic model of *Cis*-acting elements distribution pattern in 10 *CsDGATs* gene promoter regions. (**A**) These *cis*-acting elements were classified into four groups: stress-responsive, light-responsive, phytohormone-responsive, and plant-responsive, as shown in the heatmap. (**B**) The total count of these four categories is displayed in the bar plot.

**Figure 5 ijms-24-04078-f005:**
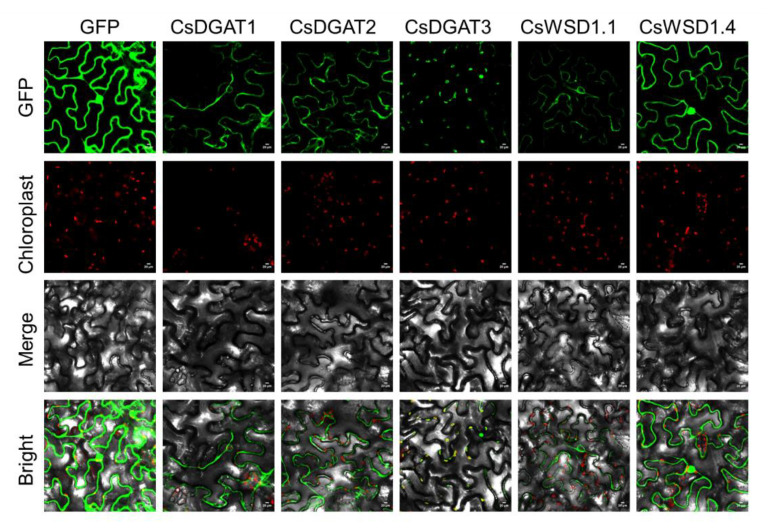
Subcellular localization analysis of CsDGATs proteins. GFP blank vector (control); Tobacco (*Nicotiana benthamiana*) leaves transiently expressed CsDGATs-GFP fusion proteins were observed through the laser scanning confocal microscope. The scale bar label in the lower right corner of each picture represents 20 μm.

**Figure 6 ijms-24-04078-f006:**
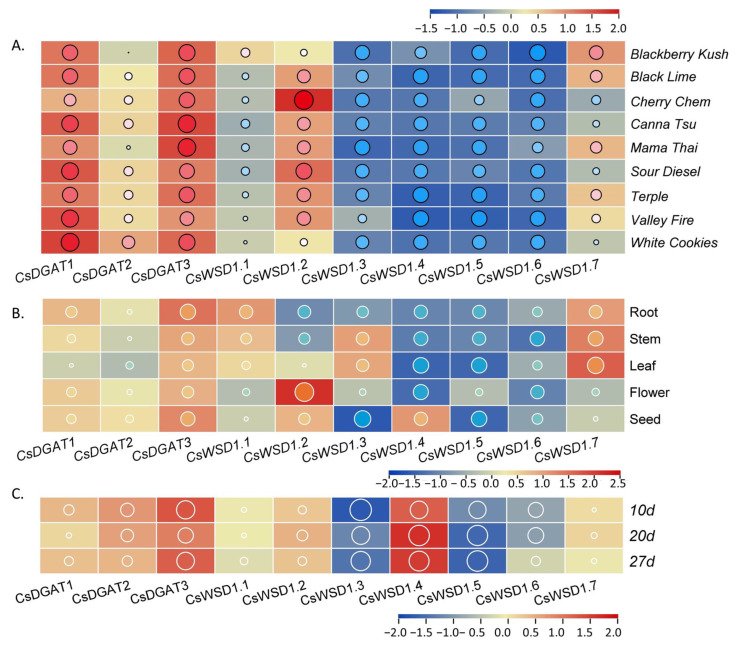
Normalized expression profiles of *CsDGAT* family genes in *C. sativa*. (**A**) Normalized expression profiles of *CsDGAT* genes in female inflorescences of ten hemp varieties based on transcriptome expression data. (**B**) Normalized expression profiles of *CsDGAT* genes in different tissues of identically hemp varieties based on transcriptome expression data. Each column represents a different tissue of hemp. (**C**) Normalized expression profiles of *CsDGAT* genes in developing seeds at different growth stages after fertilization based on transcriptome expression data. Each column represents a different growth stage of hemp. The size of circles with different colors represents the value of expression quantity. The bottom bar indicates high to low normalized expression data (red to blue).

**Figure 7 ijms-24-04078-f007:**
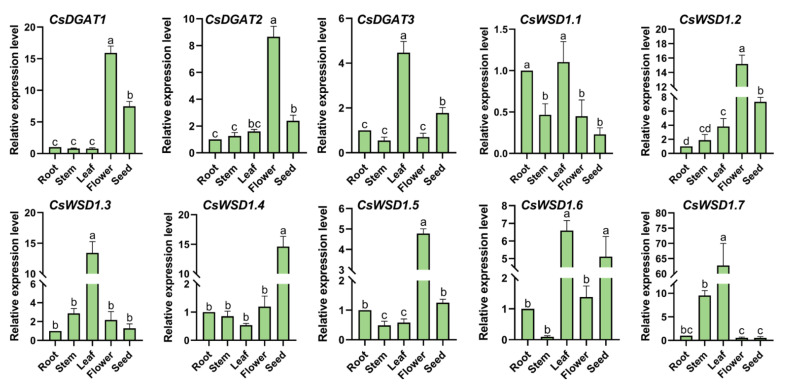
Expression patterns of the *CsDGAT* genes in different tissues detected by RT-qPCR. The characters on the *X*-axis show the different tissues of *C. sativa* (root, stem, leaf, flower, and seed). The *Y*-axis indicates the relative expression level of different genes. Three replicates were performed, and the vertical bar is the standard error. The actin7 gene was used as an internal reference. Significant differences between the different group are indicated by different letters a–d (*p* < 0.05).

**Figure 8 ijms-24-04078-f008:**
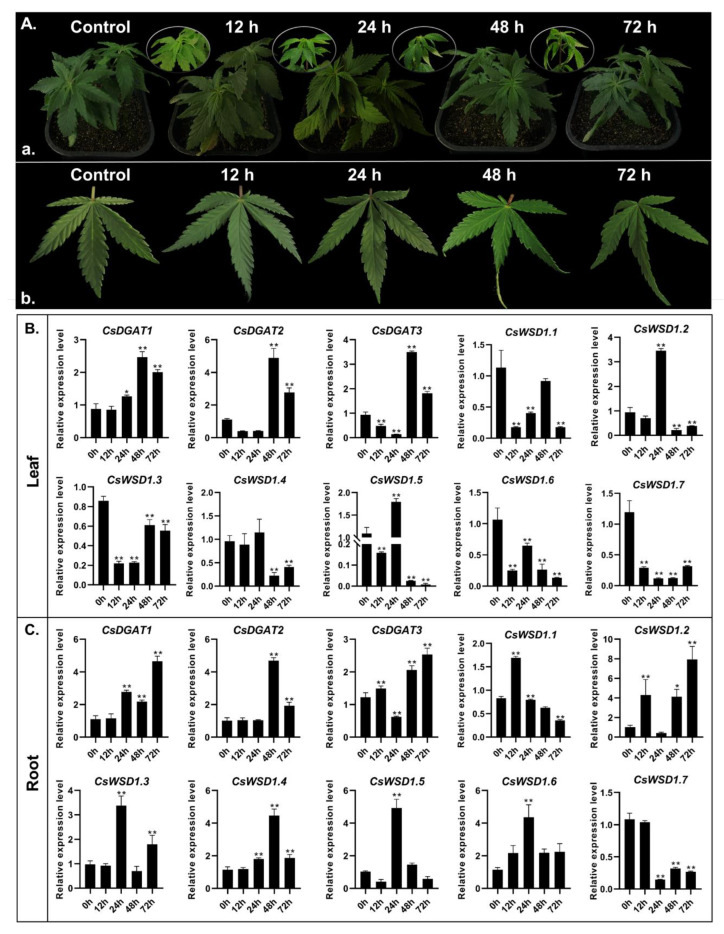
Phenotype and *CsDGAT* expression pattern analysis under cold stress response in *C. sativa*. (**A**) Phenotype analysis under cold stress in *C. sativa*. a. Phenotypic changes of hemp plants under cold stress; b. Phenotypic changes of hemp leaves under cold stress.(**B**) Expression patterns of the *CsDGAT* genes in leaves during different cold treatment time points detected by RT-qPCR. (**C**) Expression patterns of the *CsDGAT* genes in roots during different cold treatment time points detected by RT-qPCR. The characters on the X-axis show the different time points of cold treatment (0, 12, 24, 48, and 72 h). The Y-axis indicates the relative expression level of different genes. Three replicates were performed, and the vertical bar is the standard error. The actin7 gene was used as an internal reference. Significant differences between the control group and cold treatment samples are indicated by * (*p* < 0.05) and ** (*p* < 0.01).

**Table 1 ijms-24-04078-t001:** Details of genome-wide identified *DGAT* family members in *C. sativa*.

Gene ID	Gene Name	Chr	Genomic Locus	Length	MW (kDa)	PI	Subcellular Localization
LOC115704840	*CsDGAT1*	1	39274362~39279949	547	62.40394	8.7	ER, PM
LOC115703123	*CsDGAT2*	X	22711~25610	327	37.14285	9.5	ER, PM
LOC115722532	*CsDGAT3*	9	5146603~5148226	345	37.16442	8.8	Ext, Nuc, Chl
LOC115717092	*CsWSD1.1*	5	3920556~3926981	496	55.36413	8.7	ER, PM, Mit
LOC115708124	*CsWSD1.2*	1	70554466~70559429	532	60.17838	8.1	ER, Nuc, PM
LOC115703623	*CsWSD1.3*	1	70583059~70591074	511	58.01673	9	ER, PM
LOC115705098	*CsWSD1.4*	1	70705987~70710373	541	60.68999	8.3	ER, Cyto, Nuc
LOC115718623	*CsWSD1.5*	2	93659708~93666194	480	53.84373	9.4	ER, PM
LOC115708250	*CsWSD1.6*	1	70619481~70624684	556	62.91437	7	ER, PM, Nuc, Mit
LOC115702949	*CsWSD1.7*	X	3412411~3419045	473	53.17878	8.7	ER, PM, Cyto, Chl

Note: ER, Endoplasmic Reticulum; PM, PlasmaMembrane; Ext, Extracellular; Nuc, Nuclear; Chl, Chloroplast; Mit, Mitochondrial; Cyto, Cytoplasmic.

## Data Availability

RNA-seq data were downloaded from the NCBI (PRJNA498707) corresponding to the female inflorescences of 9 different hemp varieties, including the Mama Thai (MT), White Cookies (WC), Canna Tsu (CT), BlackLime (BL), Temple (T), Cherry Chem (CC), BlackBerry Kush (BB), Sour Diesel (SD), and Valley Fire (VF) varieties. Transcriptomic data corresponding to the roots, stems, leaves, flowers, and seeds of the ‘Longdama No. 9’ hemp variety were generated by our team and upload to NCBI datebase (PRJNA899681). In addition, transcriptome data from hemp seeds at different developmental periods (10d, 20d, 27d) after fertilization were used to analyze the expression profiles of *CsDGAT* genes at different developmental stages of hempseeds were publicly archived on NCBI datebase (PRJNA513221).

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
