# Peer review of "Genome-Wide Identification, Classification, and Expression Analyses of the CsDGAT Gene Family in Cannabis sativa L. and Their Response to Cold Treatment"

_ijms, 2023, doi:10.3390/ijms24044078_

Round 1

Reviewer 1 Report

The authors describe that Diacylglycerol acyltransferase (DGAT) is a key functional gene in catalyzing triacylglycerol biosynthesis and plays a role as a rate-limiting enzyme. This gene is very important. The authors' pictures are also beautifully done.

A, The gene family information analysis is systematic and some of the family members were selected for localization experiments. However, the work presented in the manuscript seems to be insufficient, I hope that the authors can add a model species (Arabidopsis, oleaginous or yeast) for overexpression of genetic transformation, which is not too difficult and has a short experimental period, so that the systematization of the research work will be beneficial.

B. In addition, according to the author's abstract and the introduction, more attention should be paid to the high content triglycerides tissues to carry out, such as increasing the work of expression analysis of seed growth and developmental stages, the work on the response to abiotic stresses should be auxiliary.

C. Alternatively, the authors could add work related to DGAT genes and abiotic stresses or highlight work related to the effects of low temperature on Hemp in the introduction. This would make the full text more logical. Also, add pictures of phenotypic changes in leaves and roots (or whole plants) at different treatment time points under cold stress conditions.

The authors could choose one of comments A, B, or C for improvement. Then the other two suggestions to add analysis and outlook in the discussion section.

Other than that, there are still some issues that need to be revised.

1. Title: If the authors motif the above comments to describe the biological story more system, the title should be modified accordingly, such as writing directly about the cold resistance or oil synthesis function of one gene member, Rather than expression analysis.

2. Introduction section: the authors should add the introduction of gene function-related research work, because the mining of functional genes is the key to the research work. Further condense the introduction of the biological function of functional components.

3Results section: All quantitative expression analysis pictures are not labeled with significant differences a,b,c, etc.

4、In Figure 1, the labeling of elliptical Cluster is out of the regional range, which can be considered by drawing the outer circle (corresponding to different colors) to indicate.

5、Figures 1 and 2 involve other species, please add the description of other species abbreviations in the figure captions.

6、The scale bar label in Figure 5 is too small, you can add a description in the caption or modify it in the picture.

Author Response

Jan 20, 2023

Dear Reviewers:

Re: Revisions to Journal of International Journal of Molecular Sciences No. ijms-2148618 by Yan et al.

We sincerely appreciate your efforts in dealing with this manuscript, and herein submit a revised version of our entitled manuscript "Genome-Wide Identification, Classification, and Expression Analyses of the CsDGAT Gene Family in Cannabis sativa L" by Bowei Yan and co-authors for review and publication in Journal of International Journal of Molecular Sciences.

We would like to thank the reviewers’ helpful advices and have revised the manuscript accordingly and carefully. We respond to the Reviewers’ Comments in the attachment file.

The revised manuscript and A list of changes have been submitted online.

Again, I would like to appreciate the reviewer for your time and efforts in reviewing this manuscript. I hope we have revised the manuscript to your satisfaction, and look forward to hearing from you soon.

Sincerely yours,

Bowei Yan, first author

E-mail address: yanbowei21@outlook.com

Corresponding author. Liguo Zhang

E-mail address: paper-special@outlook.com

Heilongjiang Academy of Agricultural Sciences, No 368, Xuefu Road, Nangang District, Harbin, Heilongjiang Province, P.R.China

Post code: 150000

Tel: +86-187-4593-4580 (M)

Reviewer 2 Report

It is a very complete job, i enjoy to read it. I only have two comments: 

Line 40. you repeated… which 70-80% are PUFAs more than 70%-80% is composed of PUFAs

Reference 2 does not say that there is between 70-80% PUFA, maybe it could be suggested but it is not written in the text. They mention around 70%.

Author Response

(The authors gave the same response as above.)

Reviewer 3 Report

In the current MS, Yan et al. analyzed the genomic architecture for different isoforms of the four classes of hemp DGAT genes in comparison to those of other plant species to improve the yield and quality of hemp seed oil. In addition, the gene structures, conserved domains, transmembrane domains, chromosomal positioning, and collinearity, phylogenetic associations, subcellular localization, and cis-acting elements associated with these CsDGATs were characterized. Moreover, the expression of the identified CsDGATs in different tissues and in response to cold stress conditions was evaluated through both qPCR and transcriptomic approaches. Although the topic is attractive, there are some concerns that should be addressed.

-Generally, the manuscript is well organized but there are some typographical and grammatical errors.

-The paper title is well stated, it is informative and concise.

-Abstract is well structured.

-The introduction was not well written, and it is too briefly presenting the subject and research problem.

L 36: First, introduce cannabis and its application. my suggestion: “Hemp (Cannabis sativa L.) is an annual herbaceous plant that has been widely used due to its industrial (10.3906/bot-1907-15), ornamental (https://doi.org/10.3390/plants11182383), nutritional (https://doi.org/10.3390/plants11233330) and pharmaceutical (https://doi.org/10.1016/j.biotechadv.2022.108074) potentials.”

-The results obtained in this study are interesting. Results are presented correctly.

-In general, the discussion was not well written. This part should be improved.

- Material and research methods are presented appropriately. The experimental setup and the description in the methods section are well structured, and the statistical analysis is correctly performed.

-Conclusion was not well written. It is too short. This part should be improved. Future studies should be discussed.

Author Response

(The authors gave the same response as above.)

Round 2

Reviewer 1 Report

The author has carried on the massive revision.

The author should add a subtitle to figure 8 in A-E. In addition, the phenotype of A-C appears to be inappropriate, with a containing 12h, whereas B and C are not 12h, which is very arbitrary, and A-C should both contain 12h, regardless of whether the phenotypic variation is significant or not. In addition, it is recommended that A-C be integrated into a single graph (Fig. 8a) , as A-C are all the same subject matter and belong to the plant phenotype, and B and C are just different perspectives and tissue closeups.

Author Response

Feb 01, 2023

Dear Reviewers:

Re: Revisions to Journal of International Journal of Molecular Sciences No. ijms-2148618 by Yan et al.

We sincerely appreciate your efforts in dealing with this manuscript, and herein submit a revised version of our entitled manuscript "Genome-Wide Identification, Classification, and Expression Analyses of the CsDGAT Gene Family in Cannabis sativa L" by Bowei Yan and co-authors for review and publication in Journal of International Journal of Molecular Sciences.

We would like to thank the reviewers’ helpful advices and have revised the manuscript accordingly and carefully. We respond to the Reviewers’ Comments in order below:

Point 1: The author should add a subtitle to figure 8 in A-E. In addition, the phenotype of A-C appears to be inappropriate, with a containing 12h, whereas B and C are not 12h, which is very arbitrary, and A-C should both contain 12h, regardless of whether the phenotypic variation is significant or not. In addition, it is recommended that A-C be integrated into a single graph (Fig. 8a) , as A-C are all the same subject matter and belong to the plant phenotype, and B and C are just different perspectives and tissue closeups.

Response 1: Thank you for your nice comments, and pointing out our problems. These opinions help us to improve acadwmic rigor of our manuscript. Based on your suggestion and request, we have made corrected modifications on the revised manuscript. Firstly, a subtitle to figure 8 has been added in the figure captions. Secondly, the phenotype picture of 12 h under cold treatment has been added to Figure 8A. Thirdly, figure A-C has been integrated into a single graph.

The revised manuscript have been submitted online.

Again, I would like to thank you for your time and efforts in reviewing this manuscript. I hope we have revised the manuscript to your satisfaction, and look forward to hearing from you soon.

Sincerely yours,

Bowei Yan, first author

E-mail address: yanbowei21@outlook.com

Corresponding author. Liguo Zhang

E-mail address: paper-special@outlook.com

Heilongjiang Academy of Agricultural Sciences, No 368, Xuefu Road, Nangang District, Harbin, Heilongjiang Province, P.R.China

Post code: 150086

Tel: +86-187-4593-4580 (M)

Reviewer 3 Report

All the comments have beed addressed. I think that the current form of the MS can be published in IJMS.

Author Response

Feb 01, 2023

Dear Reviewers:

Re: Revisions to Journal of International Journal of Molecular Sciences No. ijms-2148618 by Yan et al.

On behalf of my co-authors, we would like to thank you very much for giving us an opportunity to published our manuscript. We appreciate you very much for your positive and constructive comments and suggestions on our manuscript entitled "Genome-Wide Identification, Classification, and Expression Analyses of the CsDGAT Gene Family in Cannabis sativa L ijms-2148618". Meanwhile, we are very encouraged by your kindness and professional opinions. We will continue to work hard and strive for better results!

Sincerely yours,

Bowei Yan, first author

E-mail address: yanbowei21@outlook.com

Corresponding author. Liguo Zhang

E-mail address: paper-special@outlook.com

Heilongjiang Academy of Agricultural Sciences, No 368, Xuefu Road, Nangang District, Harbin, Heilongjiang Province, P.R.China

Post code: 150000

Tel: +86-187-4593-4580 (M)